# *Mycobacterium bovis* Tuberculosis in Two Goat Farms in Multi-Host Ecosystems in Sicily (Italy): Epidemiological, Diagnostic, and Regulatory Considerations

**DOI:** 10.3390/pathogens11060649

**Published:** 2022-06-04

**Authors:** Vincenzo Di Marco Lo Presti, Maria Teresa Capucchio, Michele Fiasconaro, Roberto Puleio, Francesco La Mancusa, Giovanna Romeo, Carmelinda Biondo, Dorotea Ippolito, Franco Guarda, Flavia Pruiti Ciarello

**Affiliations:** 1Istituto Zooprofilattico Sperimentale “A.Mirri” of Sicily, Via G. Marinuzzi 3, 90129 Palermo, Italy; vincenzo.dimarco@izssicilia.it (V.D.M.L.P.); michele.fiasconaro@izssicilia.it (M.F.); roberto.puleio@izssicilia.it (R.P.); giovanna.romeo@izssicilia.it (G.R.); carmelinda.biondo@izssicilia.it (C.B.); pruitiflavia@outlook.it (F.P.C.); 2Department of Veterinary Sciences, University of Turin, Largo P. Braccini 2, 10095 Torino, Italy; mariateresa.capucchio@unito.it (M.T.C.); guarda.franco@gmail.com (F.G.); 3Azienda Sanitaria Provinciale di Catania, Via San Paolo 5, 95123 Catania, Italy; francesco.lamancusa@aspct.it

**Keywords:** tuberculosis, goat, multi-host ecosystem, immunohistochemistry, pathology, *Mycobacterium bovis*

## Abstract

*Mycobacterium bovis* (*M. bovis*) is the causative agent of animal tuberculosis (bTB), infecting and causing disease in several animal species. In areas where there are complex interactions between reservoir hosts and susceptible species, the control of this pathogen is a challenge. The authors report two outbreaks of goat tuberculosis caused by *M. bovis* in multi-host ecosystems within two protected natural areas of Sicily, where TB is historically endemic. The first outbreak (Farm A) was identified after the incidental detection at the slaughterhouse of TB-like lesions in goat viscera ready to be disposed. Single intradermal cervical tuberculin test (SICT) was performed in Farm A on 205 goats, resulting positive in 10 (4.9%). After slaughtering, six out of ten animals showed TB-like lesions, from which *M. bovis* spoligotype SB0841 was isolated. The typing did not reveal any epidemiological connection with the neighboring cattle, suggesting that free-ranging type of management exposed the affected goat livestock or wildlife infected with other strains. The second outbreak (Farm B) was detected in a mixed farm (bovine, caprine, and ovine), where relapsing outbreaks of TB in cattle were registered in the previous years after performing the SICT in cohabiting goats. SICT resulted positive in 6/153 (3.9%), and two animals showed bTB-like lesions. No mycobacteria were cultured, and the final diagnosis of TB was achieved by histopathology and immunohistochemistry. The reported outbreaks highlight the importance of assessing the epidemiological, diagnostic, and regulatory critical issue, which is fundamental to optimizing the strategies of eradicating TB in the endemic multi-host ecosystem described.

## 1. Introduction

Animal tuberculosis (TB) is a zoonotic disease caused by microorganisms belonging to the *Mycobacterium tuberculosis* complex (MTBC). Among the members of MTBC, *M. bovis* shows the broadest host spectrum, including both domestic (cattle, goats, sheep, pigs, horses, etc.) and wild (wild boar, fallow deer, wild pigs, etc.) species [1,2]. Consequently, in multi-host ecosystems, *M. bovis* finds a single multi-species host community, and [3] its control and surveillance represent a real challenge, as the complex epidemiological dynamics are less predictable than a single-host disease [4]. For this reason, in several countries, the spread of *M. bovis* in animal species other than bovine (target host) has hampered the eradication of TB [2]. A similar context has been widely described in Sicily (southern Italy). The disease has been reported in the island in many non-bovine species, such as wild pig, wild boar, fallow deer, and sheep [5,6,7,8,9]. The wide diversity of shared strains has demonstrated the active circulation of *M. bovis* among species [9,10]. Despite decades of efforts and a general decreasing trend (from 4.41% in 2016 to 0.82% in 2021), TB prevalence in cattle is not uniform, and clusters of infection are still common. Therefore, it is reasonable to suppose that other species play a crucial role in maintaining the disease in some areas of Sicily, such as the protected natural parks, where the most common type of farming is multi-species and free roaming. Herds are often transhumant, and domestic and wild animals share shelters, common pastures, and watering points. Thus, the complex spatio-temporal interactions between cattle and other domestic and wild reservoir species are relevant risk factors for the maintenance of the disease. However, the TB reservoir role in the island has been investigated and suggested only for the free-roaming Nebrodi black pig population [6]. On the contrary, the role of goats in the disease’s dynamics has long been underestimated, as goats are classically considered less susceptible to the disease [11,12]. Nevertheless, the number of reports of TB caused by *M. bovis* in goats has significantly increased over the last decade worldwide, especially in non-officially TB-free European countries, such as the United Kingdom, Portugal, Ireland, Spain, and Italy [12,13,14,15]. In most documented outbreaks, the co-occurrence with infected cattle seems to be the common denominator [14,15,16]. There is some evidence on the reservoir role of this species in particular multi-host epidemiological systems [17]. Therefore, it is reasonable to suppose that goats may play a role in the *M. bovis* transmission, spread, and maintenance in Sicily; thus, TB control measures cannot be limited to the sole cattle but should include small ruminants especially in endemic multi-host ecosystems. However, according to the Italian law in force, TB control in other species is foreseen only if the infection in these species can compromise the outcome of the eradication plan (Art. 16 D.M. 592/94). Therefore, the lack of official epidemiological data on goat TB impairs the knowledge of the role of this species, the strains circulation, and the outbreaks traceability and of the dynamics of intra and interspecies transmission, which are essential for an effective application of the eradication plan. The present study documents the occasional finding of TB outbreaks in two goat farms located in two protected natural areas of Sicily, namely the Nebrodi Park and Madonie Park, where the disease is historically endemic. The detection of two TB outbreaks in goats represents an indicator of the presence of the disease in minor species, raising awareness of the need to activate a surveillance plan to be performed intra-vitam in the farms and postmortem at the slaughterhouse. The critical issues inherent in the epidemiological, diagnostic, and regulatory aspects for TB control in small ruminants in multi-host ecosystems will also be discussed. An analysis of the eradication plan’s scientific, managerial, and operational strategies will be provided to propose potential measures adapted to the peculiarities and specificities of multi-host ecosystems in TB-endemic territories.

## 2. Results

### 2.1. Description of the Outbreaks

The TB outbreaks occurred in two farms located in two neighboring natural areas of northeastern Sicily, the Nebrodi and the Madonie Natural Parks (Figure 1). The first outbreak occurred in a mixed sheep and goats herd located in the municipality of Ucria (Messina) in the Nebrodi Natural Park. At the time of the outbreak, the herd consisted of 335 goats and 89 sheep reared in an extensive system which was permanent for some animals and transhumant for others. Annual transhumance was carried out in southeast Sicily. Feeding and watering were based on exploiting and shifting natural pastures and natural water resources. The farm mainly produced meat and milk. Both goats and sheep were officially free of brucellosis but never tested for TB. The staff of the Istituto Zooprofilattico Sperimentale of Sicily (IZSSI)—Barcelona Area P.G. (Messina) suspected TB in May 2014 during the routine slaughtering after the accidental detection of multiple coalescing granulomatous lesions in a liver already placed in disposal tanks for its destruction. The epidemiological investigation showed that within 1 km, there were three cattle farms with which the herd of goats could have had direct or indirect contact. Two of the three cattle farms recorded TB outbreaks in the past. In particular, one farm registered an outbreak in 2015, while the second farm had several relapsing outbreaks between 2010 and 2019.

The second outbreak occurred in a mixed cattle/goat herd (Farm B) located in the municipality of Sclafani Bagni (Palermo), in the Madonie Natural Park (Figure 1). The farm was a large mixed farm established in the 1980s in which cattle, goats, sheep, and horses were bred in an extensive and semi-extensive system (Figure 2), which extends for about 700 ha on a hilly area at 300 m above the sea level. Animals were mainly fed on farm pastures, with seasonal shifting and straw pasture and feed integration. The water supply consisted of an artificial water-collection basin located within the farm. Goats were mainly bred for meat and milk production. The type of cattle breeding was the cow–calf line. Cattle were officially free from brucellosis and bovine enzootic leucosis, whereas the last TB outbreak was registered in 2014, and the farms were declared officially free from TB in 2015. The sheep and goats reared in the same farm were officially free from brucellosis and were never tested for TB. In Farm B, TB was suspected in goats after detecting a TB outbreak in cohabiting cattle.

### 2.2. Intradermotuberculin Test

In Farm A, SICT was performed in 205 animals out of 335 (61%). In Farm B, SICT was performed in 155/206 goats (54%) and 146/188 cattle (77%). Sixteen SICT reactors were detected among the tested goats: 10/205 (4.9%) in Farm A and 6/153 (3.9%) in Farm B.

### 2.3. Gross Pathology

All the SICT-positive animals were slaughtered and underwent careful postmortem examination. Visible TB-like lesions (VL) were detected in 6 out of the 10 SICT-positive goats in Farm A. Three animals showed generalized TB at different evolutive stages. Animal 1 and Animal 2 showed VLs in the lungs and annexed lymph nodes (LNs), varying in size and at different degrees of calcification (Figure 3). Moreover, the liver and spleen were affected in Animal 1, with dramatic gross alteration (Figure 4a–c and Figure 5). In addition, Animal 2 showed caseous granulomas involving prescapular, hepatic, and mammary LNs (Figure 6). The third goat (Animal 3) displayed tuberculous lesions in the parotid, prescapular, and mammary LNs. The latter showed disseminated calcifications (Figure 7a,b). VLs limited to the head and the thoracic LNs were evident in the last three goats from Farm A (Animals 4–6).

In farm B, two out of the six SICT reactors (33%) showed VLs. One goat (Animal 7) showed generalized lesions involving tonsils, liver and thoracic (mediastinal and left tracheo-bronchial) LNs (Figure 8). All the lesions were small (2–3 mm in diameter), encapsulated with a caseous calcific center. (Figure 8b) The left tracheo-bronchial LN displayed encapsulated lesions with a moderate degree of calcification (Figure 8a). In the second goat (Animal 8), granulomas were limited to the liver. In both animals, the hepatic lesions consisted of subcapsular confluent granulomas with moderate calcification near the ileal region (Figure 8c).

### 2.4. Bacteriological and Molecular Investigations

All the organs showing VLs collected from goats of Farm A revealed the presence of *M. bovis.* The typing performed at the National Reference Center for Tuberculosis, IZS of Lombardia and Emilia Romagna (CRN-TB-IZSLER) confirmed the presence of *M. bovis*, spoligotype SB0841. The molecular epidemiological investigations showed no epidemiological correlation with the outbreaks registered in the neighboring cattle farms, as the *M. bovis* spoligotype isolated was SB0120. None of the samples collected in Farm B were positive at the microbiological examination. Due to the confirmed negativity, the samples were submitted to histopathological examination.

### 2.5. Histopathologic Findings

All the culture-negative target organs (lymph nodes of the head, thorax and abdomen, tonsils, and lungs) from Farm B goats showed the presence of differently sized granulomas. In the samples without VLs, histopathology did not detect any microscopic alteration. However, a microscopic granuloma was detected in a mesenteric LN in the absence of VL in Animal 1. Histologically, lesions showed large necrotic areas surrounded by a variable number of neutrophils, epithelioid cells, giant cells, macrophages, lymphocytes, and peripheral fibroplasia (Figure 9 and Figure 10), with minimal central calcifications. These lesions were classified as grade III, according to Wango et al. (2005) and García-Jiménez et al. (2013) [18,19]. Zihel–Neelsen staining confirmed the presence of acid fast bacilli (AFB) in the observed granulomas. Moreover, immunohistochemical (IHC) staining showed a fine granular cytoplasmic staining in epithelioid cells and focal staining in some macrophages outside the granulomas (Figure 11).

## 3. Discussion

Animal tuberculosis is a zoonotic disease with important implications for both animal and human health and severe economic consequences for animal trade worldwide. The TB outbreaks in goats reported in the present study occurred in two typical multi-host ecosystems historically endemic for TB in cattle, namely the Nebrodi and Madonie natural parks in Sicily. The disease in these areas has been detected in several species such as domestic and feral pigs, fallow deer, sheep, and wild boar [5,6,7,8,9]. Thus, the two TB outbreaks described in the present study suggest that goats can acquire the infection and might have a role in the epidemiology of TB, as ascertained in other similar multi-host contexts [17]. However, the Sicilian official data on TB prevalence in small ruminants re lacking except for sporadic reports in sheep [5]. These data are essential for risk assessment and preliminary to TB-control strategies development in multi-host ecosystems where TB is endemic. The effective control of a multi-host pathogen is possible only if each species’ importance and epidemiological role in a multi-host ecosystem are known. However, despite TB being historically endemic in these territories in cattle, the surveillance in minor species lacks effectiveness both in the farm and at the slaughterhouse.

The outbreaks described highlight the challenges in TB detection in minor species and the related scientific, managerial, and operational criticalities that hamper the effectiveness of the TB eradication plan in these territories.

According to the EU Reg 625/17, ovine and caprine organs and LNs should undergo the sole visual inspection, limiting the inspection cut “when there are indications of a possible risk to human health, animal health or animal welfare”. The current approach might be sufficient in a single-host environment, where *M. bovis* infection is primarily affecting cattle and where minor species have a limited role in the disease’s dynamics. Conversely, in multi-host ecosystems where goats are exposed to the main risk factors for TB infection, this approach might be suboptimal, hampering the success of the eradication campaign. As the slaughterhouse represents the key epidemiological observatory for TB, the systematic inspective cuts of all the target organs and LNs should be performed in all species, as provided for cattle. Both the reported cases were at high risk of being undiagnosed by the routine diagnostic procedures carried out at the slaughterhouse and in the farm as laid out by the current regulation in force. The affected liver coming from Farm A was ready to be disposed without any further diagnostic investigation. Moreover, the lesions could be consistent with other bacterial infections (*Staphylococcus aureus*, *Corynebacterium pseudotuberculosis*, *Streptococcus pyogenes*, etc.), for which no specific measures are envisaged. Additionally, the affected goats from Farm B showed small lesions (<cm) limited to liver, tracheo-bronchial, and mediastinal lymph nodes, which were likely to be unnoticed by a routine meat-inspection process.

Furthermore, the present cases raise the criticalities related to the intravitam TB control in minor species within multi-host ecosystems. In Italy, TB control relies on a test and cull strategy targeting cattle, whereas other susceptible species can be tested if their presence within the herd can impair the eradication plan. Although TB control in goats is provided in specific circumstances (e.g., herds intended for the production of raw milk and in case of movement between states), targeted measures are rarely applied to species other than bovines. In the reported cases, SICT confirmed the infection in 4.9% (Farm A) and 3.9% (Farm B) of the tested animals. Additionally, the distribution and characterization of the lesions suggested an advanced stage of airborne-acquired TB, which spread to other body districts via the lymph–hematogenic route. Histopathology confirmed the advanced stage, detecting stage III granulomas with minimal calcification. The high prevalence and the severity of the lesions suggest that the disease had likely persisted in the herd for years. Therefore, a thorough TB surveillance in goats should be aimed at the earliest diagnosis possible and thus be mandatory and not left to the discretion of the veterinary service. Many European countries have already adapted their eradication campaign to multi-host ecosystems. For example, SICT is mandatorily performed in mixed goat/cattle herds in Spain and in goats epidemiologically related to cattle [17]. Similarly, in England and Scotland, the Animal and Plant Health Agency (APHA) may decide to carry out official controls in the same epidemiological conditions. APHA also allows “private” control at the request of the breeder.

The reported cases also highlight the usefulness of a multi-step diagnostic approach to improve the success of TB detection, as already tested in other species and settings [20,21,22]. Although *M. bovis* isolation is considered the gold-standard confirmatory method for TB diagnosis, culture can lack sensitivity [23]. This limitation is mainly related to analytical issues (homogenization, decontamination, culture medium, etc.), to the sample characteristics (e.g., extent of necrosis and calcification), and to the bacterial load [23,24]. Culture sensitivity appears to be around 80%, and the parallel use of solid and liquid culture media can improve it [25,26]. The unsuccessful isolation of *M. bovis* in samples collected in Farm B confirmed the limited sensitivity of this technique. In these cases, the observed lesions were small, encapsulated, and non-exudative, showing a moderate degree of calcification. However, due to the positive reaction to SICT, histopathology, ZN staining and IHC were performed, as provided by the current Italian legislation (DM 592/95), confirming the infection. Histopathology and specific TB staining also have the additional advantage of producing faster results than bacterial culture. Moreover, the microscopic assessment allows to date the infection and to characterize the underlying immune response [18,27]. The lesions detected in our case were graded as stage III granulomata according to Wangoo et al. [18] and confirmed as paucibacillary lesions. This evidence might explain the inability to culture *M. bovis*, where bacteriology might fail if necrosis is present and if the bacterial load is insufficient [23]. Unfortunately, although the infection was confirmed, the culture failure did not allow in this case to perform any further molecular analysis aimed at the epidemiological connection of the outbreak to other TB cases in the neighboring farms.

Conversely, spoligotyping was successfully performed on isolates obtained from Farm A. However, the neighboring cattle farms were not confirmed as source of the infection, as the strain isolated (SB0841) differed from the cattle one (SB0120). However, the area is characterized by a wide variety of spoligotypes [9,10]. Thus, excluding an inter-species transmission from livestock from other herds or wildlife is not possible. Both spoligotypes are the most widespread in Sicily, and in the Nebrodi area, SB0120 is the most prevalent in cattle [9,10], while SB0841 is responsible for most of the outbreaks in pigs [6]. The analysis of strain circulation is a powerful tool for elucidating the transmission dynamics both intra and interspecies, allowing outbreaks traceability [3,28]. The complex spatio-temporal interactions between susceptible species favor the transmission, spread, and persistence of TB. In these conditions, the area becomes a single epidemiological unit, where strains can circulate among a variegate multi-species host basin without any barrier. The two farms examined in the present study can be considered the prototype of Sicilian farms in natural areas. The most common type of farming in the island is multi-species and free-ranging, accounting for about 35% of the total number of farms. Several animal species (cattle, small ruminants, pigs, equids, etc.) from different farms and often with different health statuses graze together in common public pastures. This type of management makes even single-species herds mixed herds. The transhumance additionally worsens the mixture of susceptible hosts, as groups of animals are seasonally moved to different common pastures according to age, production category and reproductive stage. Moreover, the most common type of breeding is the free-range “cow–calf” line for cattle and the scarcely specialized mixed-production (meat/milk) farms for goats and sheep. These types of farms rear long-living breeding animals (>10 years), which are more likely to develop chronic forms of TB and spread tuberculous bacilli in the environment for more extended periods. Identification methods and timing represent another critical issue. Identification devices (ear tag, endoruminal bolus) are often fraudulently replaced, making the sanitary controls challenging to be thoroughly applied, especially if free-ranging animals from the same farm but reared in different sites are tested in different periods. Moreover, another risk factor is the absence of adequate biosecurity measures for free-range systems, such as fences avoiding contact with *M. bovis*-susceptible livestock and wildlife. The direct and continuous connections in time and space between hosts enable *M. bovis* to diffuse and to be maintained among the community of susceptible hosts. Therefore, although the molecular epidemiology did not find any straightforward connection with the neighboring outbreaks, it is not possible to exclude that the abovementioned factors (mixed-species pastures and watering sites, transhumance, contact with wildlife, etc.) have been crucial players in the occurrence of the goat outbreaks described. In particular, spoligotype SB0841 is diffusely present in the populations of wild or domestic free-ranging pigs, which have been suggested as TB-maintenance hosts in the area [6].

In conclusion, the goat outbreaks described gave the chance to discuss some of the critical aspects of the diagnosis and the regulation concerning TB in minor species reared in multi-host ecosystems. The first cases (Farm A) were at high risk of being undiagnosed by the routine meat inspections at the slaughterhouse. Similarly, the second outbreak (Farm B) could have gone unnoticed if diagnostic tests would not have been carried out on goats cohabiting with infected cattle (which, according to Italian law, is not mandatory but left to the discretion of the official veterinary service). The cases show that the control programs in force might be appropriate for intensive or more rational types of farming but must be implemented and adapted to multi-host archaic farming systems. Moreover, TB is a complex disease, for which the diagnosis is not always straightforward but passes through three connected and interdependent phases: (i) the intravitam diagnosis in the farm, (ii) the inspection at the slaughterhouse, and (iii) the confirmatory tests carried out in the laboratory. Each phase needs to be methodically performed and requires careful evaluation case by case by the official veterinary service. Therefore, the thorough application of the eradication plan in multi-host ecosystems must also include goats and rely on all the diagnostic tools available.

## 4. Materials and Methods

### 4.1. Anamnesis, Clinical, and Epidemiological Investigations

After formulating a reasonable suspicion of the presence of TB in the goats of both farms, several operational inspections were conducted on the farms and at the slaughterhouse, during which the anamnestic and clinical information were collected and evaluated in the diagnostic process. According to the legislation in force (Art. 16 DM 592/1995), the official veterinary service of Messina province for Farm A and Palermo for Farm B performed the investigations in collaboration with the staff of the “territorial assistance” section of the IZSS-Barcelona Area P.G (Messina) in both farms in order to confirm the infection and to establish its origin.

### 4.2. Single Intradermal Cervical Test (SICT)

The single intradermal cervical test (SICT) was performed on all animals older than 6 weeks in both Farms A and B. After measuring the skin fold basal thickness by a caliper, the test was performed, injecting intradermally bovine purified protein derivative (B-PPD, 50,000 UI produced by the Istituto Zooprofilattico of Umbria and Marche–Perugia) into the left mid-cervical region. The test was performed in the shoulder region if the animal wore any collar or bell. The thickness of the skin fold was measured after 72 h, and the results were interpreted according to the European Union Directive 97/12/EC, as positive, >4 mm increase of thickness and/or presence of one or more clinical signs such as edema, exudation, necrosis, and pain; inconclusive, 2–4 mm increase of thickness, without clinical signs; negative, <2 mm increase of thickness without clinical signs. The reactors were culled, as required by current legislation in force (D.M. 592 of 15 December 1995, chapter V—special provisions-art. 16).

### 4.3. Post-Mortem Examination

After slaughter, all the carcasses of SICT-positive animals were submitted to a careful visive inspection. Target organs (lymph nodes of the head, thorax and abdomen, tonsils, and lungs) were systematically sampled regardless of visible lesions (VL), while other organs were sampled if gross lesions were detected. After the IZSSI laboratories’ accurate examination, samples of target organs were collected and stored at −20 °C for microbiological analysis. An additional aliquot was 10% buffered formalin fixed and sent to the pathology laboratory of the IZSSI for histopathological investigations.

### 4.4. Bacteriological and Molecular Examinations

According to the manufacturer’s instructions, samples were processed in a liquid culture system BACTEC-MGIT 960 (Becton Dickinson^®^, Sparks, MD, USA). Briefly, after homogenization, samples were decontaminated using NaOH 1 N and then neutralized. The sediment was then incubated in liquid medium MGIT with PANTA antibiotics at 37 °C ± 2 for 6 weeks. When positivity was detected, smeared samples were submitted to Ziehl–Neelsen staining for mycobacterial growth confirmation. Consequently, all the positive samples were sent to the CRN-TB, IZSLER (Brescia, Italy) for confirmation and typing. Genotyping of the isolates was performed by spoligotyping [29].

### 4.5. Histology and Immunohistochemistry

Formalin-fixed and paraffin-embedded tissues were sectioned to obtain 4 μm thickness slices stained with hematoxylin and eosin. Additional sections were also stained by Ziehl–Neelsen and observed at 400× magnification to identify the presence of acid-fast bacilli (AFBs) within the lesions. Immunohistochemical investigations were performed on selected tissues. Briefly, after antigen retrieval in sodium citrate solution (pH 6.0) at 96 °C for 20 min, endogenous peroxidase activity was quenched with 3% hydrogen peroxide in methanol for 30 min. Then, the slides were treated with 1% bovine serum albumin (BSA) (Sigma-Aldrich, Saint Luis, MO, USA) for 30 min and incubated for 1 h at room temperature in 0.1% BSA with a polyclonal rabbit antibody against *M. bovis* (BIORBYT, ORB 100411) and diluted 1:100 in 0.01 M PBS. Sections were then treated for 30 min with secondary biotinylated immunoglobulin anti-rabbit antibody (DAKO, LSAB Kit, K0690, Glostrup, Denmark) and incubated with a streptavidin-horseradish peroxidase conjugate for 1 h, followed by chromogen 3-3′ diaminobenzidine tetrahydrochloride treatment for 1 min. A counterstaining with Mayer’s hematoxylin was also done. Positive and negative (additional slide with omission of the primary antibody) controls were also used. Images of stained slides were captured by Leica (Wetzlar, Germany) DMR microscope equipped with a DS-Fi1, Nikon (Tokyo, Japan) digital camera. The classification of the granuloma was based on cell composition, caseous necrosis, and mineralization according to Wangoo et al. and García-Jiménez et al., 2013 [18,19].

## Figures and Tables

**Figure 1 pathogens-11-00649-f001:**
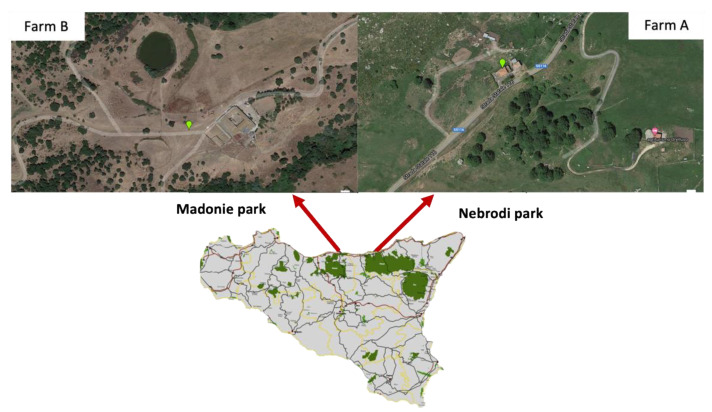
Geolocation of the two Farms A and B in the Nebrodi and Madonie Parks, respectively.

**Figure 2 pathogens-11-00649-f002:**
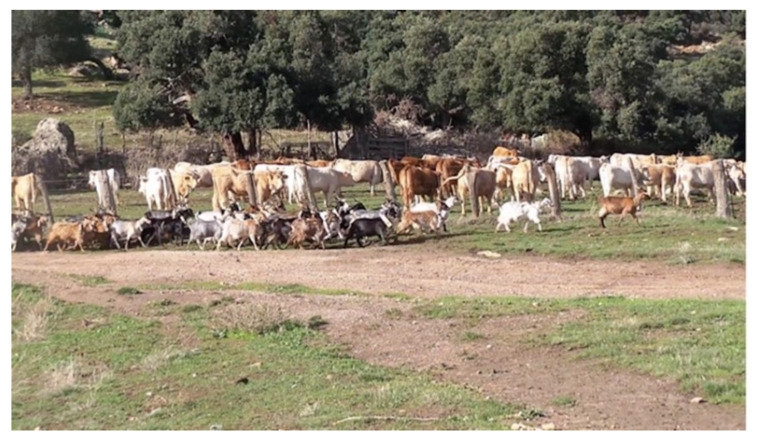
Farm B—Typical mixed herd of goat and cattle in the Madonie Park.

**Figure 3 pathogens-11-00649-f003:**
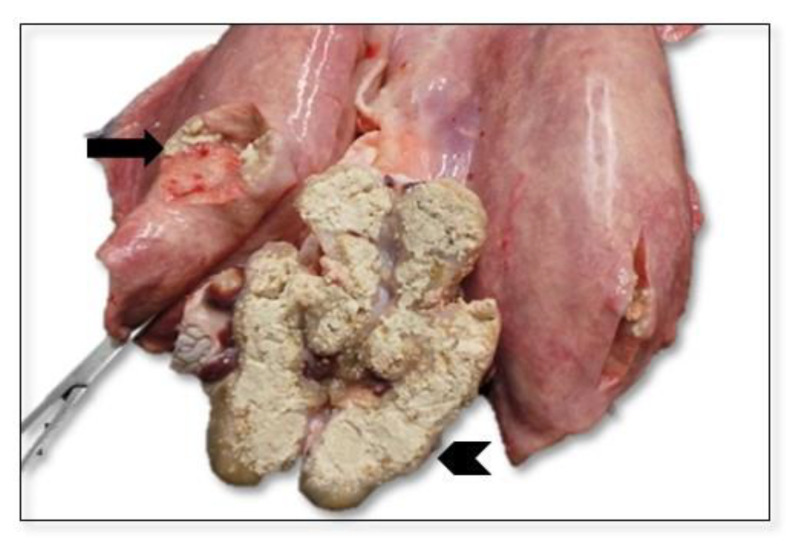
Farm A Goat—Animal 1. Lungs and tributary lymph nodes (tracheo-bronchial and mediastinal). Lungs with multiple subpleural granulomas varying in size with a firm, elastic consistency and caseous necrosis (arrow). Mediastinal LN was enlarged and strident when cut, whitish in color, and cheesy and calcified in appearance (arrowhead).

**Figure 4 pathogens-11-00649-f004:**
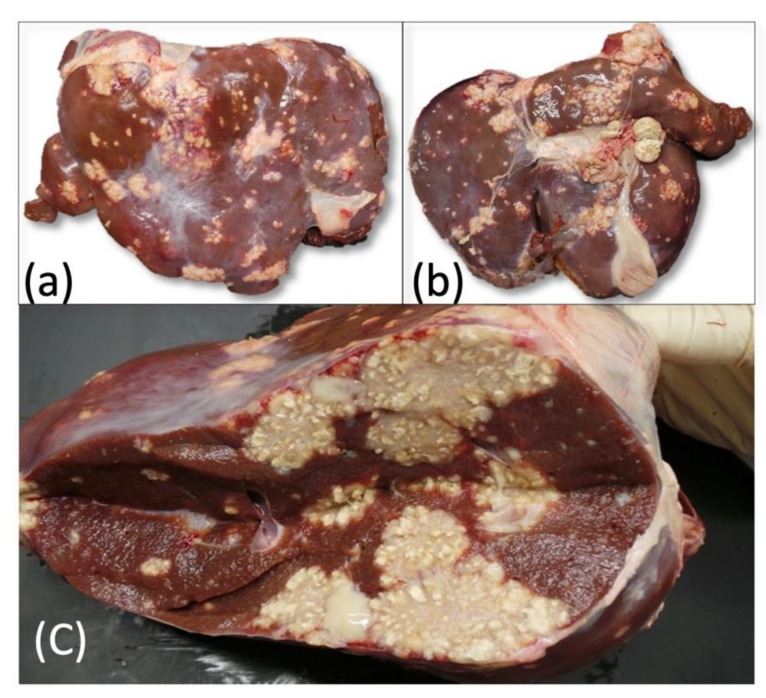
Farm A Goat—Animal 1. Liver with multifocal granulomatous hepatitis. Multiple, irregularly shaped tuberculous foci, randomly distributed, on the external (**a**,**b**) and on the cut surfaces (**c**), with involvement of hepatic lymph-nodes (**b**).

**Figure 5 pathogens-11-00649-f005:**
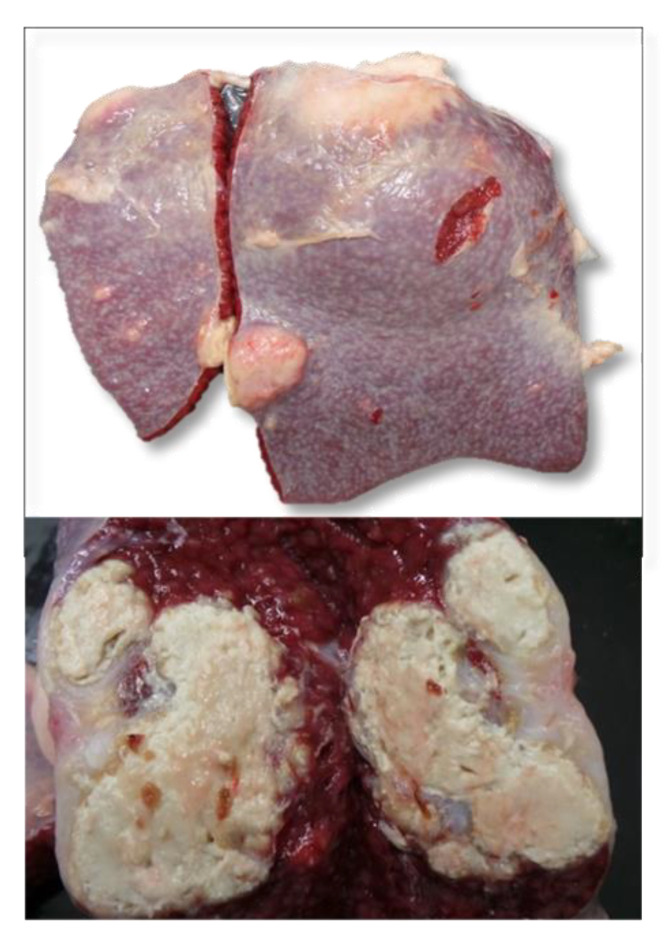
Farm A Goat—Animal 1. Spleen displaying disseminated voluminous granulomas characterized by severe caseous necrosis.

**Figure 6 pathogens-11-00649-f006:**
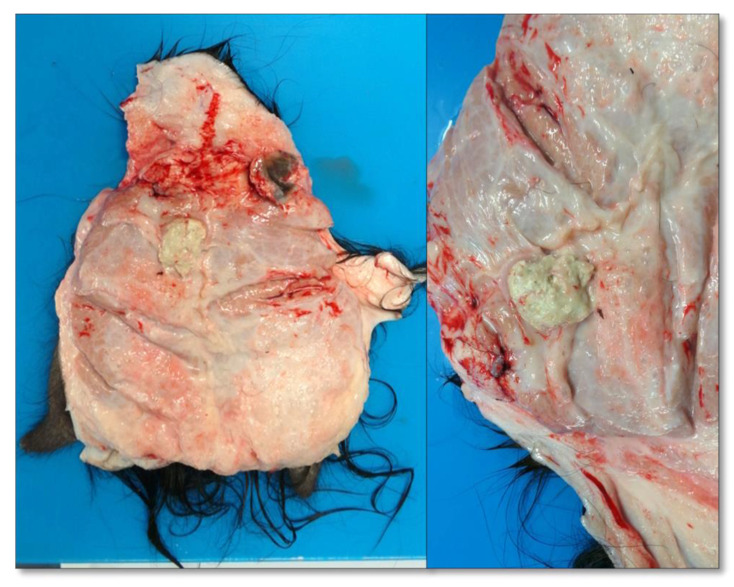
Farm A Goat—Animal 2. Mammary lymph nodes with caseous tubercular lesions.

**Figure 7 pathogens-11-00649-f007:**
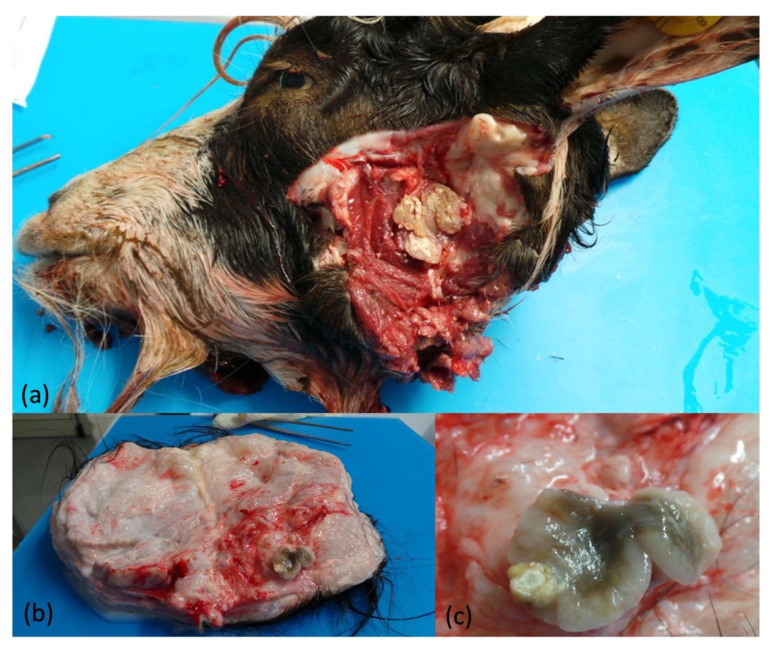
Farm A Goat—Animal 3. Parotid lymph nodes with caseous not calcified exudate (**a**). Mammary lymph node, with a caseous calcified granuloma (**b**). Detail from the mammary lymph node (**c**).

**Figure 8 pathogens-11-00649-f008:**
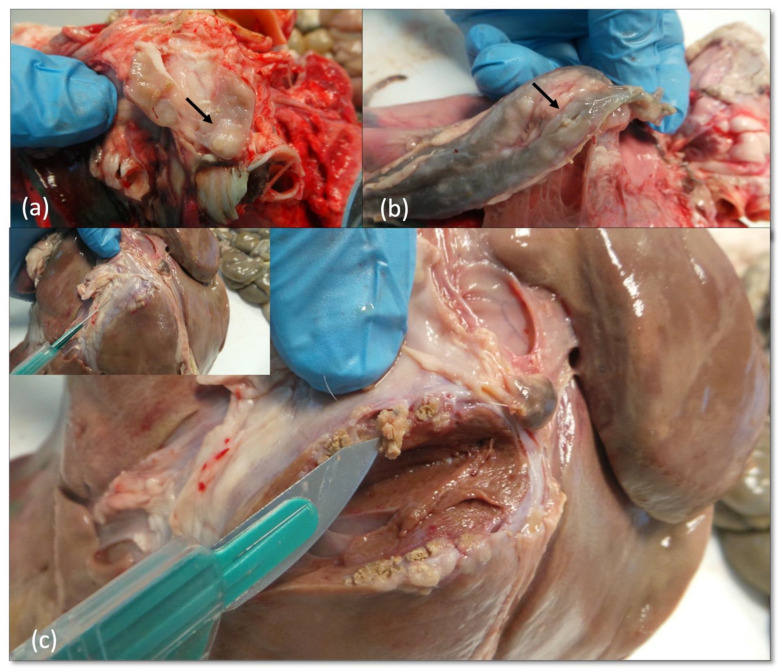
Farm B Goat—Animal 7. Left bronchial (**a**) and mediastinal (**b**) LNs with small, encapsulated granulomas (arrow). Impairment of hepatic parenchyma due to the presence of sub/capsular coalescing granulomatous lesions with a moderate degree of calcification (**c**).

**Figure 9 pathogens-11-00649-f009:**
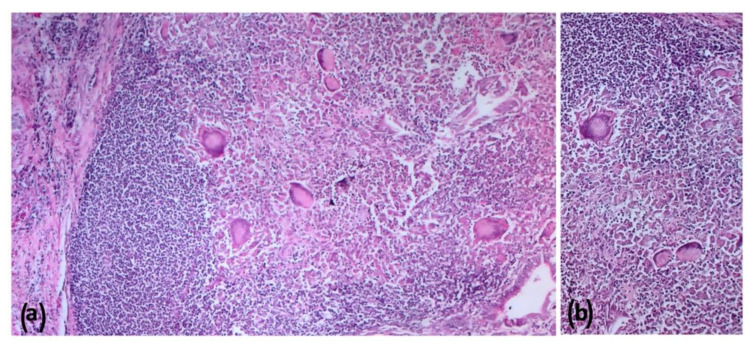
Farm B Goat—Liver. Coalescing granulomas composed by a variable number of giant cells, epithelioid cells, macrophages, lymphocytes, and peripheral fibroplasia; hematoxilin and eosin (HE); 100* (**a**); B: higher magnification of the mononuclear HE, 200* (**b**).

**Figure 10 pathogens-11-00649-f010:**
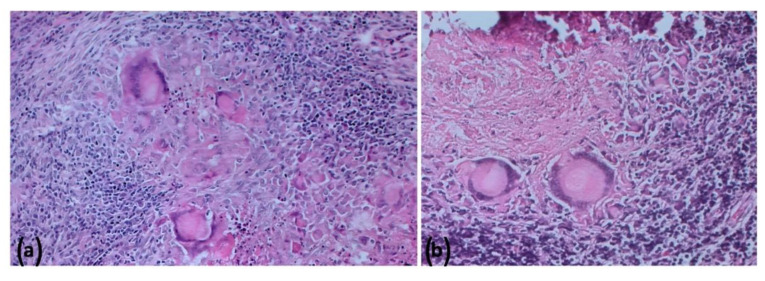
Farm B Goat—Mediastinal lymph node. Voluminous granulomas with numerous giant cells surrounded by a variable number of epithelioid cells, macrophages, and lymphocytes; hematoxylin and eosin (HE); 200* (**a**); B: higher magnification of the granuloma: at the top the caseous necrotic center with calcifications surrounded by the mononuclear infiltrate with voluminous giant cells. HE, 400* (**b**).

**Figure 11 pathogens-11-00649-f011:**
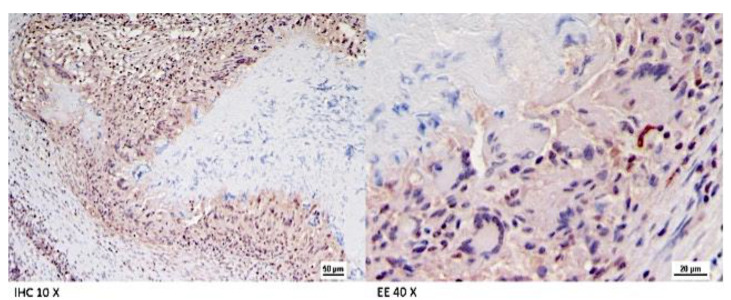
Farm B Goat—Fine granular brown staining in the cytoplasm of epithelioid cells and multinucleated giant cells. Immunohistochemistry against *M. bovis*, HE counterstain, 400*.

## Data Availability

Data sharing not applicable. No new data were created or analyzed in this study.

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
