# Peer review of "Mycobacterium bovis Tuberculosis in Two Goat Farms in Multi-Host Ecosystems in Sicily (Italy): Epidemiological, Diagnostic, and Regulatory Considerations"

_pathogens, 2022, doi:10.3390/pathogens11060649_

Round 1
Reviewer 1 Report
This study describes two outbreaks of tuberculosis in goats at two farms in Sicily. M. bovis was confirmed in goats at farm A, but the spoligotype differed to that previously found in the neighbouring cattle. M. bovis was not confirmed in goats by bacteriology at farm B so it was not possible to establish an epidemiological link with the cattle.
The revision has greatly improved the manuscript and some of the concerns raised have been addressed. I don’t have any major issues with the study design or interpretation of data. Perhaps the Abstract should include a line stating that no direct epidemiological link was established between the outbreaks in the goats and associated cattle herds. Including this statement will not detract from the theme of paper, if anything it will strengthen the paper.
I’m still not convinced that long sections on regulatory considerations are merited. In line 325 it is stated that the outbreaks give the chance to discuss critical aspects of the diagnosis and regulation concerning TB etc. I can see the logic in discussing the diagnostic issues but not so clear as to why it needs to digress into a whole range of regulatory issues. The study is strong enough to retain its main focus, in my opinion much of rest is distracting and unnecessary. I am not saying to remove it all, but it needs to be shortened substantially for the reader who is primarily interested in the description of goat TB. Among the sections that could be reduced by 50% or more, include Discussion lines 221-262 and 298-317. This would make for a much better Discussion section overall.
Line 212: Should be ‘except for sporadic..etc’.
Line 264: Delete ‘want to’.
Line 281: Replace ‘the unsucess of the bacterial culture, which..)’ with ‘the inability to culture M. bovis, where bacteriology might fail if necrosis..etc.
=
Author Response
Reviewer’s comment: This study describes two outbreaks of tuberculosis in goats at two farms in Sicily. M. bovis was confirmed in goats at farm A, but the spoligotype differed to that previously found in the neighbouring cattle. M. bovis was not confirmed in goats by bacteriology at farm B so it was not possible to establish an epidemiological link with the cattle.
The revision has greatly improved the manuscript and some of the concerns raised have been addressed. I don’t have any major issues with the study design or interpretation of data. Perhaps the Abstract should include a line stating that no direct epidemiological link was established between the outbreaks in the goats and associated cattle herds. Including this statement will not detract from the theme of paper, if anything it will strengthen the paper.
Response to the reviewer: We are glad that all the main concerns have been successfully address. We agree that the abstract should include the statement suggested; therefore, it was added as follows:
“The typing did not reveal any epidemiological connection with the neighbouring cattle, suggesting that free-ranging type of management exposed the affected goats live-stock or wildlife infected with other strains.”
Reviewer’s comment: I’m still not convinced that long sections on regulatory considerations are merited. In line 325 it is stated that the outbreaks give the chance to discuss critical aspects of the diagnosis and regulation concerning TB etc. I can see the logic in discussing the diagnostic issues but not so clear as to why it needs to digress into a whole range of regulatory issues. The study is strong enough to retain its main focus, in my opinion much of rest is distracting and unnecessary. I am not saying to remove it all, but it needs to be shortened substantially for the reader who is primarily interested in the description of goat TB. Among the sections that could be reduced by 50% or more, include Discussion lines 221-262 and 298-317. This would make for a much better Discussion section overall.
Response to the reviewer: The paragraphs mentioned were shortened according to your suggestions. However, in order to maintain the focus of our discussion, we have kept most of the regulatory considerations already drafted. Indeed, the manuscript’s main purpose is to raise the main criticalities related to TB control in goats, which are both diagnostic and regulatory. In our opinion, diagnostic issues cannot be separated from the legislator ones. However, we do believe that addressing the legislation gaps regarding multi-host free-ranging farms is crucial, and that was one of the main aims of the present manuscript. Therefore we think that the regulation in force should be revised and adapted to these peculiar epidemiological settings. TB has persisted for decades and still persists in many territories, and in some Italian regions, such as Sicily, one of the main reasons for the failure of the eradication plan is related to the discretionary of the controls of minor species at the farm and to the inspection at the slaughterhouse, which does not provide thorough investigation for other reservoir species, and especially for goats.
Reviewer’s comment:
Line 212: Should be ‘except for sporadic..etc’.
Line 264: Delete ‘want to’.
Line 281: Replace ‘the unsucess of the bacterial culture, which..)’ with ‘the inability to culture M. bovis, where bacteriology might fail if necrosis..etc
Response to reviewer: all the suggestions have been addressed
Reviewer 2 Report
After the reviewers suggestions were made, the work only gained in value.
Author Response
Many thanks for your help in the revision process.
We made our best efforts to address all your valuable comments and suggestions.
Thanks also for the positive response.
This manuscript is a resubmission of an earlier submission. The following is a list of the peer review reports and author responses from that submission.
Round 1
Reviewer 1 Report
This study describes two outbreaks of goat tuberculosis caused by M. bovis (SB0841 spoligotype) in a multi-host ecosystem within two protected natural areas of Sicily, where TB is historically endemic. The infection in goats discovered during the rutine slaughtering and confirmed by histopathology, mycobacterial culture and strain identification. Overall, the manuscript is very interesting and will enrich knowledge on the prevalence of TB in goats.
I agree with the authors’opinion the slaughterhouse represents the key epidemiological observatory for TB, especially in multi-host areas. A systematic inspection should be performed in all animals coming from multi/host TB endemic areas, performing the systematic inspective cuts of all the target lymph nodes and target organs as in cattle.
I don't have many comments. The work is well written, it is a pleasure and curious to read. All sections contain relevant information. In addition, the work is enriched with numerous figures that add to its attractiveness.
Specific comments
Line 23. spolygotype – should be spoligotype
The same error appears several more times throughout the text
Line 36. Mycobacterium tuberculosis (Mycobacterium in capital letters and Mycobacterium tuberculosis in italics) complex (MTBC). This is the first time I meet for M. tuberculosis complex to have the acronym MTC.
Line 188. Expand the shortcut: CRN-TB-IZSLER
Line 188-191. The molecular epidemiological investigations showed no…. This sentence should be in the Discussion section when discussing the results obtained.
Line 236. [20] - without bold
Suggestion:
In the future, if the authors have a herd of small ruminants suspected of having TB, it would be good to take blood from these animals and use it for further research. The results of the Bovigam test could be compared with the SICT, or even other serological tests could be used to improve them. Ante mortem diagnosis of animal TB (especially of species other than cattle) is limping and this blood serum would be a great material for detecting specific antybodies.
Reviewer 2 Report
In this work, the authors have studied the prevalence of Mycobacterial infection among goat population from two different farmlands in Italy. Development of multi-drug resistant tuberculosis is a global threat as over 1 million people die of tuberculosis every year. The transmission of tuberculosis between different alternative hosts acts a breeding ground for emergence of multidrug resistance. Hence, careful monitoring of these zoonosis events like transmission of pathogen causing tuberculosis in cattle to goat is one of the key priorities for public health officials. In this work, authors have reported cases from two farmlands where the goats have been infected with tuberculosis. The authors have suggested that the goats have acquired the pathogen from the cattle population due to free mixing among these two species of grazing mammals. The authors have provided histopathological proof of acute form of tuberculosis in the goat population as marked by the stage III granuloma with minimal calcification.
However, the authors failed to any strong molecular evidence that the goats in both the farms have the same strain of Mycobacterium as the cattle nearby. Also, the authors have not conducted any experiment where they kept infected cattle with a healthy goat in any enclose space to show that the goat gets infected by the cattle. Moreover, no specimen was collected from the second farm. Overall, the paper lacks rigorous scientific experiments to conclude that zoonosis happened between cattle’s and goats. Altogether, there is lack of any scientific evidence conclude this is an act of zoonosis. The author should design clear scientific experiments as a supporting proof of transmission of the same bacterial strain from cattle to goat. Taken together, the data is preliminary and not conclusive.
Reviewer 3 Report
This study describes two outbreaks of tuberculosis in goats at two farms in Sicily. M. bovis was confirmed in goats at farm A, but the spoligotype differed to that previously found in the neighbouring cattle. M. bovis was not confirmed in goats by bacteriology at farm B so it was not possible to establish an epidemiological link with the cattle.
I think this is a fair summary of the highlights of the paper. In this context and in my opinion the manuscript is far too long for the information and results that are conveyed; it could be reduced by 50% and still hold all of the relevant information.
The Abstract and Introduction sections are fine and do set the scene appropriately.
In section 2.1, the description of the outbreaks could be shortened substantially (e.g., is it important to the study that farm A is owned by three brothers, etc? See lines 92). The same applies for the description of Farm B.
The Results section contains many graphic images of lesions, the vast majority are not necessary to show (maybe Fig 4, should be retained). Most of the information in the gross pathology section should be put into a Table, summarising the organs and lymph nodes where lesions were found. A general description of the lesions can then be provided, rather than a detailed description of each lesion type. Likewise, for the description of the histopathology results.
The Discussion section is too long and lacks focus (e.g., lines 224-264, 275-299, 322-335). As no epidemiological links were clearly established between the goats and cattle at both farms, much of this section is not necessary. I would keep the key elements that retain the focus on the goat TB.
There are some specific points that require discussion. In line 265 and elsewhere, it is pointed out that the majority of goats identified with the skin test had generalised TB. This does raise questions about the potency of the tuberculin used in the skin test, as it appears to preferentially detect advanced disease and has apparent low sensitivity for non-lesioned animals. How many infected goats do the authors think were missed by the skin test in these herds? I note that the tuberculin was produced locally (line 364); is this an OIE approved tuberculin? Was the potency of the tuberculin determined according the guidelines in the OIE manual? All of this merits discussion.
It is puzzling that the infection was not confirmed in the goats in farm B given the extent of the lesions? I note in lines 307-310 that some explanations have been given. I read (lines 384) that 1N NaOH was used to decontaminate samples: in many circumstances this can be particularly harsh and reduce the bacterial load considerably.
Lines 119: Should be ‘mainly’.
Lines 140: Replace ‘remnants with ‘remaining’.
=